# Mechanistic Insights into the Successful Development of Combination Therapy of Enfortumab Vedotin and Pembrolizumab for the Treatment of Locally Advanced or Metastatic Urothelial Cancer

**DOI:** 10.3390/cancers16173071

**Published:** 2024-09-04

**Authors:** Caroline Taylor, Kamai M. Patterson, Devira Friedman, Silvia M. Bacot, Gerald M. Feldman, Tao Wang

**Affiliations:** Office of Pharmaceutical Quality Research, Center for Drug Evaluation and Research, Food and Drug Administration, Silver Spring, MD 20993, USA; caroline.taylor@fda.hhs.gov (C.T.); devira.friedman@fda.hhs.gov (D.F.); silvia.bacot@fda.hhs.gov (S.M.B.)

**Keywords:** antibody–drug conjugate, immune checkpoint inhibitors, enfortumab vedotin, pembrolizumab, anticancer combination therapy, urothelial carcinoma

## Abstract

**Simple Summary:**

Antibody–drug conjugates (ADCs) and immune checkpoint inhibitors (ICIs) are two promising therapeutic modalities against many types of cancers. However, many patients develop resistance. The resistance mechanisms to ADCs and ICIs have been comprehensively illuminated in this review. A combination of ADCs and ICIs has been explored to overcome resistance to ADC or ICI single treatment. Recently, a clinical study demonstrated that a combination of enfortumab vedotin (EV), an ADC against Nectin-4, with the ICI pembrolizumab achieves remarkable clinical efficacy as the first-line therapy for the treatment of locally advanced or metastatic urothelial carcinoma. The underlying mechanism is likely due to the enhancement of pembrolizumab-induced anticancer immunity mediated by EV. With the emerging use of combination therapy strategy, it is critical to understand the mechanism of successful and/or failed clinical studies for the future development of combination therapy of ADCs with ICIs.

**Abstract:**

Antibody–drug conjugates (ADCs) consist of an antibody backbone that recognizes and binds to a target antigen expressed on tumor cells and a small molecule chemotherapy payload that is conjugated to the antibody via a linker. ADCs are one of the most promising therapeutic modalities for the treatment of various cancers. However, many patients have developed resistance to this form of therapy. Extensive efforts have been dedicated to identifying an effective combination of ADCs with other types of anticancer therapies to potentially overcome this resistance. A recent clinical study demonstrated that a combination of the ADC enfortumab vedotin (EV) with the immune checkpoint inhibitor (ICI) pembrolizumab can achieve remarkable clinical efficacy as the first-line therapy for the treatment of locally advanced or metastatic urothelial carcinoma (la/mUC)—leading to the first approval of a combination therapy of an ADC with an ICI for the treatment of cancer patients. In this review, we highlight knowledge and understanding gained from the successful development of EV and the combination therapy of EV with ICI for the treatment of la/mUC. Using urothelial carcinoma as an example, we will focus on dissecting the underlying mechanisms necessary for the development of this type of combination therapy for a variety of cancers.

## 1. Introduction

Since the approval of the first antibody–drug conjugate (ADC) gemtuzumab ozogamicin (GO; Mylotarg) for the treatment of acute myeloid leukemia in 2000 (withdrawn in 2010, reapproved in 2017) [1,2], 13 ADCs have been approved by the United States Food and Drug Administration (FDA) and more than a hundred new ADCs are currently being tested in active clinical trials for the treatment of various hematopoietic and solid cancers [3]. Despite the bumpy start of GO notwithstanding, ADCs have emerged as one of the most promising anticancer therapeutic modalities, due to their remarkable clinical efficacy and relative safety profile compared to chemotherapy [3,4,5,6,7]. ADC therapeutics combine the advantages of the precise molecular targeting capability of an antibody with the added potency of the small-molecule drug payload at local concentrations that would otherwise not be tolerated when used alone. After specifically binding to a tumor antigen expressed on the surface of tumor cells, ADCs are internalized by the tumor cells and subsequently processed in the lysosomes where the payload is then released within the tumor cells, leading to cell death. Additionally, ADCs have been shown to exert a bystander effect, in which the payloads are released from the targeted cells into the tumor microenvironment leading to the death of neighboring tumor cells expressing negative or low levels of the target antigen, as well as nearby stromal cells that support the survival of tumor cells [4,5,6,7].

In contrast to the promising clinical benefits, such as trastuzumab deruxtecan (T-DXd) for the treatment of HER2 (human epidermal growth factor receptor 2)-positive breast cancers and other types of HER2-positive unresectable or metastatic solid cancers [8,9,10], many FDA-approved ADCs have relatively modest efficacy and can cause severe toxicity [3,11]. Furthermore, many patients do not respond to the initial treatment with ADCs (primary resistance) or develop resistance after initial positive responses to the ADC treatment (acquired resistance) [12,13,14,15]. For example, enfortumab vedotin (EV), a fully human monoclonal ADC targeting Nectin-4-positive tumor cells that carries a payload of the microtubule-disrupting agent monomethyl auristatin E (MMAE) [16] was approved for the treatment of patients with previously treated locally advanced or metastatic urothelial carcinoma (la/mUC). However, many patients do not respond to this treatment (overall response rate 44%), and the median duration of response (DoR) is only 7.6 months [17,18]. Thus, many ADC development programs have been discontinued at different stages of clinical development due to a lack of efficacy and/or the induction of severe toxicity [6,19,20].

In the context of la/mUC, in addition to developing effective and safe ADCs by targeting novel antigens or designing new ADCs targeting Nectin-4 by employing new payloads or new backbone antibodies or developing new conjugation technologies to overcome drug resistance [21,22,23], combining ADCs with other therapeutic modalities has emerged as one of the major strategies to improve treatment efficacy. Such combination therapies include ADCs combined with nearly all other types of anticancer therapies, including chemotherapy, targeted therapy, immune checkpoint inhibitors, anticancer vaccines, and cell therapies, among others [7,24,25]. Combination therapy with two different ADCs has also been tested for the treatment of la/mUC. For example, EV with sacituzumab govitecan (SG), an ADC against trophoblast cell surface antigen 2 (TROP-2) with a SN38 payload (topoisomerase I inhibitor), demonstrated superior clinical activity for the treatment of patients with la/mUC compared to EV or SG treatment alone in a phase I trial study [26].

Although many ADC combination therapies have not succeeded in various stages of clinical trials due to an apparent lack of a synergistic effect on increased efficacy or, conversely, an apparent increase in drug toxicity compared to single treatment, a recent clinical study using EV (previously known as ASG-22CE) with pembrolizumab, an immune checkpoint inhibitor (ICI) against programmed cell death-1 (PD-1), showed remarkable clinical efficacy, in which patients receiving the combination therapy demonstrated double the rates of progression-free survival and overall survival compared to the current standard chemotherapy as the first-line therapy for the treatment of patients with la/mUC [27]. The successful trial results led to the first approval of a combination anticancer therapy of an ADC with an ICI by the FDA. In this review, using urothelial carcinoma (UC) as an example, we summarize the history of the development of EV and the combination of EV with pembrolizumab for the treatment of la/UC. We focus on illustrating the underlying mechanisms of combined ADCs with ICIs to lay the foundation for the future development of effective ADC combination therapies (Figure 1).

## 2. Development of Enfortumab Vedotin for the Treatment of la/mUC

EV is an ADC targeting Nectin-4-positive tumor cells [16]. Nectin-4, a member of the immunoglobulin superfamily, is a single-chain type I transmembrane glycoprotein consisting of three immunoglobulin-like extracellular domains, a single transmembrane domain, and a cytoplastic tail domain [28,29,30]. Nectin-4 can be cleaved by the proteases ADAM10 (a disintegrin and metalloproteinase domain containing protein 10) and ADAM17 [30,31,32,33].

Nectin-4 has relatively restricted expression in normal tissues. The transcript and protein of Nectin-4 were found to be expressed in endothelial cells inside human placenta villi via Northern blot and immunohistochemistry analyses, respectively. Nectin-4 is moderately expressed in human skin keratinocytes and other epithelial cells of the bladder, skin, salivary gland (ducts), gastrointestinal tract, and breast ducts [34,35]. Nectin-4 is also highly expressed in many types of cancers. In one study, immunohistochemical analysis of Nectin-4 was performed on 2394 various cancer patient samples from tissue arrays. It was shown that 69% of patient samples and 60% of UC patient samples are positive for Nectin-4 [16]. In data from multiple clinical trials, Nectin-4 was expressed in most tumor tissues from la/mUC patients (Nectin-4 was expressed up to 99% patients in the EV-201 trial) [36,37]. Furthermore, several studies demonstrated that the expression level of Nectin-4 is associated with poor prognosis in many types of cancers, such as UC, breast cancer, lung cancer, and ovarian cancer, among others [33,38,39]. The physiological functions of Nectin-4, however, remain unclear from the few studies that are available. Nectin-4 was initially identified as an entry receptor for measles and herpes virus via the interaction with the viral attachment protein and was suggested as a potential therapeutic target for oncolytic viral therapy [34,40,41,42]. The most well-studied function of Nectin-4 is its identification as an adhesion molecule. Nectin-4 is connected to the actin cytoskeleton by interacting with afadin at cell–matrix junctions leading to the activation of many signaling pathways, such as phosphatidylinositol 3′ 3′–kinase (PI3K)-Akt. Nectin-4 was also shown to interact with integrin αvβ3 to modulate cell adhesion, migration, cell survival, and death [28,29,30,42,43]. The binding of Nectin-4 with itself or another Nectin family member, Nectin-1, also was reported to play a role in the adhesion function of Nectin-4 [28,44]. A homozygous mutation in the Nectin-4 gene was found to be associated with human ectodermal dysplasia–syndactyly syndrome (characterized by abnormal hair, teeth, and nails, along with bilateral cutaneous syndactyly of the fingers and toes), possibly due to Nectin-4 losing its ability to interact with Nectin-1 [45].

The functionality and mechanisms of Nectin-4 in cancer development and metastasis are also not fully understood [42,46,47]. Several groups have demonstrated that Nectin-4 activates the PI3K/Akt pathway to modulate tumor cell proliferation, survival, angiogenesis, and metastasis [30,46,47]. In ovarian cancer, Boylan et al., reported that blockading the interaction between Nectin-4 and Nectin-1 using an anti-Nectin-1 antibody or anti-Nectin-4 antibody inhibited tumor cell migration. Knockdown of the Nectin-4 gene in ovarian cancer cell lines inhibited cell migration and aggregation and increased the expression of markers of epithelial-to-mesenchymal transition (EMT) [48]. The expression of Nectin-4 was also found to modulate tumor cell proliferation, angiogenesis, migration, and metastasis in other types of cancers, including UC, breast cancer, pancreatic cancer, and esophageal cancer, among others [49,50,51]. A study by Klümper et al. demonstrated that Nectin-4 expression is lower in metastatic tissues than primary tissues potentially due to the loss of adhesion in the primary tumor tissues [52]. Expression of Nectin-4 was also found to contribute to tumor cell resistance to chemotherapy in colon cancer [53].

Due to the high expression of Nectin-4 in many types of cancers and its limited expression in normal tissues, Nectin-4 was identified as a good target antigen for ADC therapy. In a preclinical study, Challita-Eid et al. [16] demonstrated that the backbone antibody of the hybridoma version EV (ASG-22M6) and CHO version EV (ASG-22CE or ASG-ME, used for the clinical study, and expressed in Chinese hamster ovary cells) could specifically bind to the V domain of human Nectin-4. In addition, EV was shown to induce potent cytotoxicity in Nectin-4-positive breast cancer and prostate cancer cell lines. Further, in vivo experiments demonstrated that EV inhibits tumor growth in mouse xenograft tumor models of bladder cancer, pancreatic cancer, breast cancer, and lung cancer, as well as in a patient-derived xenograft model (PDX). EV also eradicated established xenografts in vivo [30]. Furthermore, EV induced a bystander cytotoxicity effect to kill Nectin-4-negative tumor cells by releasing its MMAE payload in Nectin-4-overexpressing bladder cancer cell lines and induced anti-tumor immune responses [54].

Encouraged by the promising preclinical study data, a phase I clinical trial EV-101 was conducted to study the tolerability and efficacy of EV for the treatment of heavily pretreated mUC (Table 1) [55]. EV as a single agent was generally well-tolerated, and patients showed meaningful and durable responses to the therapy (Table 1) [55]. In another phase I trial tested in Japanese patients, EV showed a similar effect in la/mUC patients in terms of efficacy and the safety profile (Table 1) [36]. In a follow-up multiple center single-arm, phase II trial (EV-201), the efficacy and safety profiles of EV were further confirmed as second-line therapy for cisplatin-ineligible la/mUC patients who were previously treated with PD-1 or PD-L1 inhibitors [37]. In the pivotal randomized controlled phase III trial (EV-301), EV demonstrated superior clinical activity to the standard-of-care chemotherapy in patients with la/mUC who had previously received a platinum-containing chemotherapy and experienced disease progression during or following treatment with a PD-1/L1 inhibitor (Table 1) [17], leading to the first FDA approval of EV as a second-line therapy for la/mUC patients [18]. In the 2-year follow-up study of EV-301 trial patients, the long-term clinical efficacy of EV was observed in terms of overall survival and progression-free survival compared to standard chemotherapy [56]. As Nectin-4 is also highly expressed in prostate cancer, breast cancer, non-small cell lung cancer (NSCLC), and gastroesophageal cancers, among others, several clinical trials are currently underway to test the clinical efficacy and safety of EV for these various types of cancer [55,57,58].

In summary, Nectin-4 has been confirmed as a promising target for the second-line treatment of la/mUC. Even with the increase in overall survival (OS) and progression-free survival (PFS) in la/mUC patients, most patients eventually developed resistance to EV therapy and fewer than thirty percent of patients survived more than 2 years. Thus, the clinical activity of EV treatment alone as a first-line therapy has not been demonstrated compared to standard chemotherapy alone [60]. Therefore, the development of a strategy to overcome this resistance to EV treatment is needed for the treatment of patients with la/mUC who otherwise have limited therapeutic options.

## 3. Development of Enfortumab Vedotin and Pembrolizumab as a Combination Therapy for Patients with la/mUC

To improve the efficacy of EV for the treatment of patients with la/mUC, many clinical studies are underway to test the efficacy and safety of combined EV with other therapeutical modalities, including chemotherapy, targeted therapy, and immune therapies, such as immune checkpoint inhibitors, for the treatment of various cancers, including UC [25,61]. The most active and advanced clinical program is investigating whether a combination of EV with pembrolizumab for the treatment of patients with la/mUC can achieve better clinical outcomes in various therapeutic settings (Table 1).

Anti-PD-1 therapeutic antibodies alone, including pembrolizumab, nivolumab, cemiplimab, dostarlimab, or durvalumab, as well as the anti-PD-L1 antibodies atezolizumab and avelumab have been previously tested as single-agent therapies in clinical studies for the treatment of patients with different types of UC in different settings [62,63,64,65,66]. The results of the clinical activities of these ICIs for the treatment of UC were mixed. While pembrolizumab significantly increased overall survival compared to chemotherapy as a second-line therapy for advanced UC [67] and demonstrated efficacy for the treatment of patients with BCG (Bacillus Calmette–Guerin)-unresponsive high-grade non–muscle-invasive bladder cancer (HR NMIBC) [68], pembrolizumab alone or in combination with platinum-based chemotherapy as a first-line therapy did not significantly improve efficacy in terms of overall survival and progression-free survival compared to standard chemotherapy treatment (Table 2) [69,70]. In an adjuvant setting for the treatment of patients with muscle-invasive urothelial carcinoma (MIUC) with a high risk of recurrence, pembrolizumab significantly increased disease-free survival but not overall survival [71]. Meanwhile, on the other hand, avelumab has been demonstrated to be effective in a maintenance therapy setting for advanced or mUC (a/mUC) patients who initially responded to chemotherapy [72]. These data along with data from other ICIs indicated that the ICI alone was effective in treating some patients with various stages of UC, but the effects were generally moderate because most patients do not experience any long-term benefits of these therapies [63,64,73]. Thus, identifying a combination therapy of ICIs, such as pembrolizumab, with other therapies to increase efficacy was actively pursued in many clinical trials for the treatment of la/mUC (Table 2). One notable achievement was that nivolumab in combination with the chemotherapies cisplatin and gemcitabine significantly increased the progression-free survival and overall survival compared to chemotherapy as the first-line treatment of adult patients with unresectable UC or mUC [74].

With respect to the combination therapy of EV and pembrolizumab, the initial phase Ib/II EV-103/KEYNOTE-869 trial (NCT03288545) was conducted to test EV alone or EV in combination with pembrolizumab as a first-line therapy for patients with la/mUC who had not received prior systemic therapy and were ineligible for a standard cisplatin-containing chemotherapy. The objective response rate for this combination therapy was 68%, with complete and partial responses of 12% and 55%, respectively. Combination therapy achieved a high confirmed overall response rate (cORR) with durable responses (Table 1) [59,60,82]. No formal statistical comparisons were conducted between EV and EV plus pembrolizumab in this trial. In the pivotal randomized EV-302/KN-A39 (NCT04223856) phase III trial, the efficacy of the combination therapy as a first-line therapy for patients with la/mUC was evaluated. The median OS was 31.5 months with EV plus pembrolizumab treatment compared to 16.1 months with platinum-based chemotherapy. Median PFS was 12.5 months in the EV plus pembrolizumab treatment group compared to 6.3 months in the platinum-based chemotherapy group (Table 1) [27]. Of note, the patient enrollment criteria were different between the phase 1b/II and phase III trials. Combination therapy in phase III was used as first-line therapy regardless of whether patients were ineligible for platinum-based chemotherapy, whereas the phase 1b/II trial only recruited patient’s ineligible for platinum-based chemotherapy [27,59,60].

In summary, combination therapy of EV with pembrolizumab improves treatment efficacy compared to the current standard chemotherapy resulting in a meaningful change for the first-line treatment of patients with la/mUC. Understanding the mechanisms underlying this improved efficacy of combination therapy could benefit the identification of biomarkers and the development of new ADC combination therapies for the treatment of UC and other types of cancers.

## 4. Mechanisms of Resistance to Enfortumab Vedotin Treatment for Urothelial Carcinoma

Tumor cells develop many mechanisms of resistance to combination anticancer therapies. To understand why some patients develop resistance to combination therapy, understanding the mechanisms underlying resistance to single therapy is needed. Some mechanisms of resistance to ADC treatment were detailed in several recent reviews [7,12,13,14,15,19,83]. Major ADC resistance attributes include, but are not limited to, the alteration of target antigen expression, which impacts ADC antibody binding to the tumor cells, changes in ADC internalization and subsequent processing in the lysosome, which impacts the release of payload into cells, and direct resistance to payload drugs, which impacts the effectiveness of payload-induced tumor cell death, among others.

These same principles apply to tumor cell resistance to EV treatment [84,85,86]. The most studied mechanism has been the contribution of the expression of Nectin-4 in primary and acquired resistance to EV treatment. In both EV-101 and EV-201 clinical trials, the expression level of Nectin-4 did not appear to correlate with the patients’ responses to EV treatment [37,55], whereas other studies indicated that Nectin-4 expression could act as a biomarker to predict patients’ responses to the EV treatment. Klümper et al. conducted a series of retrospective analyses on the association of patients’ responses to EV treatment with the level of Nectin-4 expression in primary tumors and matched metastatic tissues from EV-treated UC patients. They found that the absence or low expression levels of membranous Nectin-4 was associated with a decreased response to EV treatment for UC patients [52,87]. Additionally, Nectin-4 gene amplification was associated with increased progression-free survival in mUC patients treated with EV [88,89]. Chu et al. also reported that the difference in the Nectin-4 expression level in different UC tumor subtypes dictated the patients’ responses to the EV treatment [90].

The discrepancy in the role of Nectin-4 expression in EV resistance may be due to the following: (1) differences in assays detecting Nectin-4 expression (e.g., how the tissue sample collection was performed, or the testing protocol used); (2) differences in the location of Nectin-4 expression. For example, in samples from the EV-201 trial, 97 percent of samples were positive for Nectin-4 (but whether Nectin-4 was expressed on the cell surface or intracellular (or both) was not assessed); (3) differences in baseline therapy, such as ICI or chemotherapy (which were also associated with patients’ responses to EV treatment [91]); (4) highly heterogeneous expression of Nectin-4 in different tumor subtypes. For example, Nectin-4 has been reported to be highly expressed in luminal tumors, with lower expression in basal/squamous tumors, and substantially lower or completely absent in neuroendocrine subtype tumors [90]; (5) heterogeneous expression of Nectin-4 in the same subtype of UC. In this regard, Garczyk et al. demonstrated that Nectin-4 expression in urothelial HR NMIBC was also heterogeneous [92]. Importantly, results from most of the previous studies should be interpreted cautiously because most Nectin-4 detection assays have not been properly validated, most studies use a small patient sample size, and many of these studies are retrospective in nature [93]. In addition to clinical studies, experimentally, the knockdown of Nectin-4 expression in EV-sensitive cancer cell lines decreased EV-induced cell death. Similarly, the overexpression of Nectin-4 in a Nectin-4-low-expression cell line increased sensitivity to EV treatment [48]. Collectively, these results suggest that Nectin-4 expression levels, especial membranous Nectin-4 may, at least in part, dictate a patients’ response to EV treatment in UC or other types of cancers, because membranous Nectin-4 expression may play a critical role in EV binding and the internalization of EV into tumor cells.

In terms of the mechanisms of acquired resistance, Chang et al. reported that an EV-resistant bladder cancer cell line expressed a comparable level of Nectin-4 compared to the parental cell line but was more resistant to treatment using the payload MMAE [94]. The resistance to MMAE could have been due to the increased expression of ATP-binding cassette (ABC) transporter genes that could lead to increased efflux of the payload. Supporting this, the expression of efflux pump ABC transporters was found to predict the therapeutic efficacy of EV in UC patients [95,96]. Of note, increased expression of the ABC transporter was the most common mechanism of cancer cell resistance to chemotherapy [97] and was found to contribute to cancer cell resistance to other ADCs, such as trastuzumab maytansinoid [15,98]. Moreover, blockading the expression of the multiple drug resistance protein MDR-1/p-glycoprotein (encoded by ABC1 gene) using ABC transporter inhibitors restored drug sensitivity in anti-Nectin-4 ADC-resistant breast cancer cells [86]. Other mechanisms of drug resistance, such as somatic alterations in the tumor suppression genes TP53 and MDM2, have also been reported to contribute to cancer cell resistance to EV treatment [99]. The impacts of the internalization of EV and lysosomal functions on EV resistance have not been reported.

In summary, the differences in levels of Nectin-4 expression likely contribute to primary resistance to EV treatment, but not to acquired resistance. A prospective and biomarker-driven clinical trial and standardized Nectin-4 expression assessment assays are needed to define the impacts of the Nectin-4 expression level on the effectiveness of EV treatment. Resistance to payload might be a major acquired resistance mechanism to the treatment of EV in UC patients. Many strategies have been developed to overcome resistance to the payload, such as the conjugation of multiple types of payloads to ADCs as opposed to a single payload [100,101]. Fully understanding the mechanisms of resistance is necessary for developing new strategies to overcome resistance to ADCs.

## 5. Mechanisms of Resistance to Pembrolizumab Treatment in Urothelial Carcinoma

ICIs, such as anti-programmed cell death 1 (PD-1)/PD-1 ligand 1 (PD-L1) monoclonal antibodies (mAbs) and cytotoxic T-lymphocyte-associated protein 4 (CTLA-4) mAbs, have been approved by the FDA and have demonstrated efficacy against many types of cancers. However, the efficacy of ICI monotherapy in many cancers remains modest, and most patients develop resistance to ICI therapy. The mechanisms of ICI resistance are complex and still not fully understood. Several reviews elegantly illustrated the mechanisms of resistance to ICIs for the treatment of various cancers [102,103,104,105,106,107,108], including in UC patients [64,66,109,110]. The major resistant mechanisms to ICI treatment include but are not limited to the following: first, increased expression of other immune checkpoint molecules on T cells, such as CTLA-4, PD-L2, lymphocyte-activation gene 3 (LAG3), T cell immunoglobulin, and mucin-domain containing-3 (TIM-3), and T cell immunoglobulin and ITIM domain (TIGIT), among others [106,107,111,112]. The interaction of these immune checkpoint molecules with immune checkpoint ligands impedes the anticancer T cell response. For example, the upregulation of PD-1, LAG-3, and CTLA-4 on T cells contributed to ovarian cancer resistance to ICI treatment [113]. The expression of these checkpoint molecules was also a predictive and prognostic biomarker for patients with UC [114] and was associated with patients’ responses to ICI therapy. In addition, PD-L2 appears to play a critical role in resistance to ICIs. For example, blocking PD-L2 with a soluble form of PD-1 overcomes resistance to ICI treatment in ovarian cancer [115]. More recently, the combination of nivolumab with relatlimab, an anti-LAG3 therapeutic antibody, achieved better clinical activity than nivolumab alone for untreated advanced melanoma [116,117]. Mechanistically, several recent studies demonstrated that PD-1 and LAG3 differentially regulate anticancer CD8 T effector functions by lowering the threshold of the activation of T cell receptor signaling [118,119,120].Together, these data suggest that the expression of immune checkpoint molecules modulates anticancer T cell responses and confers resistance to anti-PD-1/PD-L1 therapy.

Second is the downregulation or loss of major histocompatibility complex (MHC) molecules on tumor cells to evade T cell recognition. For example, it was reported that patients with low expression or loss of the expression of MHC developed resistance to ICI therapy due to the deficiency in the ability to present tumor neoantigens to T cells [121,122]. The restoration of the expression of MHC molecules enhanced ICI-mediated anticancer immunity [123,124].

Third is the deficiency in the activation of signaling pathways in T cells and tumor cells. The most notable pathway contributing to ICI resistance is the interferon-γ (IFN-γ) receptor-Jak1/Jak2–STAT1 pathway in tumor cells and T cells [108,125]. While the induction of IFN-γ was a major mechanism of action of ICIs, activation of the IFN-γ R-Jak1/2–STAT1 pathway in tumor cells by T cells producing IFN-γ also contributed to tumor cell resistance to ICIs. For example, Benci et al. demonstrated that prolonged IFN-γ treatment induced genetic and epigenetic changes in cancer cells and upregulated many immunosuppressive checkpoint molecules. Blocking the IFN-γ activation pathway restored tumor cell responses to the ICI treatment [125];

Finally is immunosuppressive tumor stromal cells within the tumor microenvironment. The well-studied immunosuppressive cells include tumor-associated macrophages (TAMs), myeloid-derived suppressor cells (MDSCs), and regulatory T cells (Tregs). All these cells have been demonstrated to contribute to tumor cell resistance to ICI treatment, and targeting these cells (TAMs [126,127,128], MDSCs [129,130], Tregs [131,132]) has been shown to enhance the efficacy of ICI treatment. These cells (TAMs [133], MDSCs [134,135], Treg [136]) also played a critical role in ICI-resistance mechanisms in UC patients. Several studies also demonstrated that the immune-suppressive microenvironment is a predictive biomarker for immunotherapy in NMIBC and MIBC [137,138,139,140]. Along this line, another important consideration is that the differences in tumor subtypes may contribute to the ICI response. For example, among the five subtypes of UC, only the neuronal subtype responded well to ICI therapy, while other subtypes (luminal-infiltrated, basal-squamous, luminal-papillary, and luminal) did not respond well. The difference in the ICI response was likely attributed to the heterogeneity of tumor cells and tumor microenvironment in these UC subtypes [141].

The abovementioned resistance mechanisms not only provide the rationale to develop more effective therapies but also potentially lead to identifying biomarkers to predict which patients respond to ICI therapy, such as immune checkpoint molecules and suppressive immune cells. In addition, the following two biomarkers are also critical for ICIs: (1) tumor mutational burden (TMB). TMB is defined as the number of non-inherited (somatic) mutations per million bases in the tumor’s DNA. The mutated tumor genes are potentially processed to neoantigens in the antigen presenting cells (APCs), thus dictating the formation of the antigenic peptide/MHC complexes recognized by T cells. Supporting this, it has been shown that cancer patients with a high TMB respond better to ICI therapy than patients with a low TMB in various cancers [142,143]. For example, in the phase II Keynote-158 trial, pembrolizumab was effective for the treatment of TMB-high solid tumors, leading, for the first time, to the FDA approval of a cancer treatment based on using TMB as a biomarker [144,145]. Similarly, it has been shown that a high TMB is also a predictive biomarker for UC patients treated with ICIs [64,146], as well as many other types of cancers [147]. (2) tumor-infiltrating lymphocytes (TIL). For UC, the number of TILs was a prognostic biomarker [148] and was associated with patients’ responses to ICI therapy [149]. Although the abovementioned resistance mechanisms are well recognized, data from other studies showed conflicting results. For example, it has been reported that the TMB and TILs were not associated with UC patients’ responses to ICI treatment [150], including pembrolizumab [136]. These inconsistencies are likely due to the difference in experimental models, the complexity and heterogeneity of tumor cells and the tumor microenvironment, the fact that many clinical studies are retrospective trials based on small clinical sample sizes, and that no standard assays were used for assessing experimental parameters in these studies.

Although many biomarkers have been proposed, it is critical that a prospective clinical trial with a sufficient size of enrollment needs to be conducted to validate the biomarkers for ICIs [151,152]. One example is that a phase 3 randomized trial with 700 patients has identified the expression of PD-L1, TMBs, and others as potential biomarkers for aUC patients treated with avelumab as a maintenance therapy after first-line chemotherapy [153]. For UC, a recent meta-analysis has identified that the expression of PD-L1 of PD-L1 is a prognostic biomarker for anti-PD1/PD-L1 therapy but not a predictive biomarker for efficacy [154]. Detailed information on the mechanism-driven discovery and identification of clinical biomarkers for ICI therapy, including for UC, was elegantly illuminated by several groups [138,155,156,157,158].

In summary, the mechanisms of the resistance of tumor cells to ICI therapy are mainly due to tumor cell intrinsic mechanisms, such as the formation of an antigenic/MHC complex and/or extrinsic mechanisms, such as a highly immunosuppressive tumor microenvironment leading to the suppression of anticancer T cell immune responses (Figure 2). Combining ICI with other therapies against these mechanisms could lead to more effective ICI therapy.

## 6. The Successful Development of Combined Enfortumab Vedotin with Pembrolizumab for the Treatment of Urothelial Carcinoma

In preclinical studies, EV has been shown to induce immunogenic cell death, a type of cell death that can elicit antitumor immunity, including antitumor immune memory [159,160]. EV also has been reported to synergize with ICIs to inhibit tumor growth in an animal model [54,161]. Clinically, in the EV-301 trial, durable responses were observed in UC patients treated with EV alone [56]. In addition, in the EV-103 trial, the durable responses to EV treatment were observed in patients who discontinued treatment [37,162,163]. These two clinical observations resemble typical responses found in cancer patients treated with ICIs, supporting the immunomodulatory effects of EV.

The rationales and evidence of the enhancement of ICI-mediated anticancer activity mediated by EV include, but are not limited to, the following lines of evidence: first, an HLA-A2-restricted CD8 T cell antigenic epitope is present in the Nectin-4 protein amino acid sequence. Lopez et al. found that the Nectin-4 protein contains a nine-amino-acid peptide (VLVPPLPSL) that could bind to HLA-A*02:01 molecules. Peptide-pulsed CD8 T cells lysed Nectin-4-positive breast cancer cells. In addition, CD8 T cell clones derived from breast cancer patients recognized this epitope and induced cell death in HLA-A2-positive and Nectin-4-positive breast cancer cells. Therefore, the presence of this CD8 T cell epitope in Nectin-4-positive cells potentially contributed to overcoming the tumor cell evasion of CD8 T cell-mediated cell death, thus rendering tumor cells more sensitive to ICI treatment [124,164,165]. Supporting this, a retrospective study by Ueki et al. indicated that UC patients with a higher level of Nectin-4 expression responded better to pembrolizumab than patients with lower levels of Nectin-4 expression [166].

Second is increased expression of MHC class I and II expression. EV treatment increases the expression of MHC molecules in a xenograft tumor model [161]. Activation of the PI3K signaling pathway has been shown to downregulate the expression of MHC molecules [167]. Therefore, increased MHC expression mediated by EV treatment may be due inhibition of the Nectin-4-mediated activation of the AKT pathway. Another source of increased MHC molecules could be due to the increased infiltration of APCs, such as DCs and macrophages. Enhanced recruitment of APCs was observed in a xenograft model [161] and in other ADCs, including T-DXd [168,169].

Third is low PD-L1 expression and a high TMB. In a retrospective analysis of clinical data from advanced UC patients treated with EV, longer OS was associated with low expression of PD-L1 and a high TMB [170]. Mechanistically, Nectin-4 may increase the expression of PD-L1 via the αvβ3-integrin-mediated upregulation of PD-L1 expression, rendering tumor cells more sensitive to anti-PD-1/PD-L1 treatment [171]. Therefore, additional pembrolizumab treatment may have a synergistic effect to induce stronger anticancer T cell response by blockading the Nectin-4-induced upregulation of PD-L1 and recognizing more neoantigens due to the high TMB.

Lastly is the binding of Nectin-4 to T cell immunoreceptor with Ig and ITIM domains (TIGIT). TIGIT is a potent immune checkpoint molecule expressed on T cells, NK cells, and dendritic cells [172]. While the blockade of TIGIT alone induces modest anticancer T cell responses, the combination of TIGIT blockade with other ICIs significantly inhibits tumor growth by enhancing anticancer T cell and NK cell responses in preclinical and clinical studies [173,174,175,176,177]. TIGIT was originally identified as a receptor for Nectin-2 and plays a role in modulating the activation of anticancer NK cell and T cell responses [178]. Reches et al. first identified Nectin-4 as a ligand of TIGIT. Blocking the interaction of Nectin-4 with TIGIT by using an anti-Nectin-4 antibody induces NK cell-mediated cytotoxicity in breast cancer cell lines and in a xenograft model [179]. Studies from Ganguli et al. confirmed Nectin-4 binding to TIGIT via surface plasmon resonance (SPR) analysis [180].

The results from the above studies demonstrate that Nectin-4 plays a role in modulating anticancer T cell responses; however, the effects of EV on anticancer T cell responses have not been validated using EV treatment in preclinical and clinical studies. Future work is required to address this important issue to facilitate an understanding of the mechanisms of EV treatment. EV treatment might enhance APC functions by promoting the formation of a peptide/MHC complex, enhancing the recruitment of APCs to the tumor site, and blocking immune checkpoint molecule TIGIT-mediated immunosuppressive functions.

Although many studies convincingly demonstrated that EV treatment increases pembrolizumab-mediated anticancer T cell responses, whether pembrolizumab enhances EV-mediated anticancer cell death remains unknown. One possibility is that pembrolizumab treatment induces the production of many cytokines (i.e., interferon-λ), which may increase the expression of Nectin-4, thus enhancing EV-mediated cell death. Nonetheless, it is critical to understand the contribution of pembrolizumab treatment to EV-induced cell death. In addition, there is a possibility that EV and pembrolizumab can exert anticancer effects independently. For example, a retrospective study by Tomiyama et al. indicated that there was no correlation between the expression of Nectin-4 and PD-L1 in upper-tract urothelial carcinoma (UTUC). Similar expression patterns were observed in tumor tissues from patients with advanced UC. Therefore, EV may induce cell death in Nectin-4-positive and PD-L1-negative cancer cells, while pembrolizumab mainly induces anticancer effects on PD-L1-positive and Nectin-4-negative cancer cells [181,182].

Of note, many combination therapies of ADCs with ICIs failed in the various stages of drug development due to a lack of efficacy or because they caused severe toxicity. For example, a combination of trastuzumab emtansine (T-DM1) with atezolizumab did not show a clinically meaningful improvement in progression-free survival compared to T-DXd treatment alone [183], although T-DXd, like many ADCs, has been shown to enhance ICI-mediated anticancer T cell responses through the upregulation of HLA class expression and enhancement of anti-PDL1-mAb-induced anticancer T responses in mouse preclinical models [184,185]. Importantly, many FDA-approved indications for atezolizumab treatment require PD-L1 as a biomarker, such as lung cancer [186]. Therefore, there is a possibility that patients with high PD-L1 expression may achieve meaningful clinical benefits when treated with atezolizumab with T-DM1.

In summary, many mechanisms may contribute to the enhanced efficacy of combined EV with pembrolizumab compared to single treatment, especially their ability to enhance anticancer T cell responses [187,188]. Although fully understanding these underlying mechanisms remains challenging, recent advances in clinical studies indicate that anticancer immunity is one of critical attributes for the efficacy of combination therapy of ICIs with ADCs and that combination therapy represents a promising avenue for the treatment of various cancers.

## 7. Conclusions

ICIs and ADCs have already changed the landscape of anticancer therapy. The successful development of combination therapies of EV with pembrolizumab for UC patients also changed clinical practice for la/mUC patients. With many ongoing clinical trials testing the combination therapy of ICIs and ADCs for the treatment of various cancer patients, hopefully more effective therapies will eventually emerge. The challenges lying ahead include the identification of biomarkers to predict which patients will most likely respond to the combination therapy and understanding the mechanisms of resistance that eventually interfere with the development of effective combination therapy [189]. Recent advances in clinical studies indicate that anticancer immunity is critical for the efficacy of combination therapy of ICIs with ADCs. Our evolving understanding of the mechanisms of action of how ADCs and ICIs modulate anticancer responses will ultimately provide more promising therapeutic options for cancer patients.

## Figures and Tables

**Figure 1 cancers-16-03071-f001:**
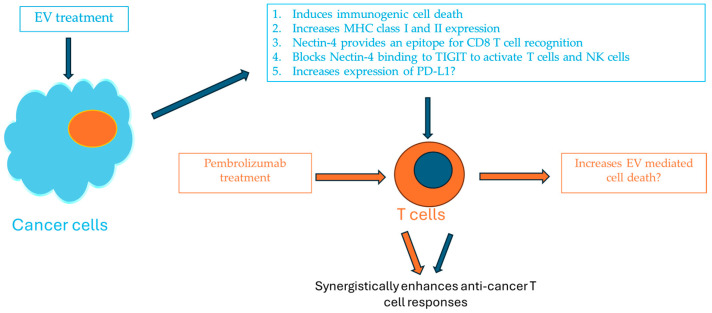
Potential mechanisms of enhanced anticancer T immune responses with the combination of pembrolizumab and enfortumab vedotin.

**Figure 2 cancers-16-03071-f002:**
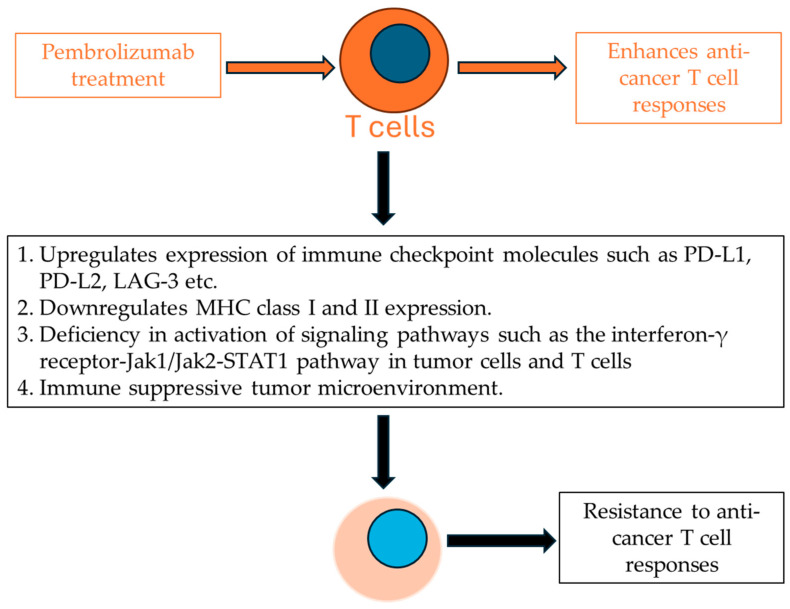
Major mechanisms of resistance to pembrolizumab treatment in UC.

**Table 1 cancers-16-03071-t001:** Clinical efficacy and safety of EV and pembrolizumab alone and in combination for the treatment of la/mUC.

Study	Regimen	Inclusion Criteria	HR (OS or PFS)	ORR (%)	DoR (months)	mOS (months)	mPFS (months)	Citation
EV101 Phase I NCT02091999	EV	≥1 L chemo. and Nectin-4-positive solid tumors, including mUC *	N/A	43	7.4	12.3	5.4	[55]
NCT03070990	EV	2 L la/mUC Dose escalation in Japanese patients with la/mUC	N/A	35.3	N/A	N/A	N/A	[36]
EV-201 Phase II NCT03219333	EV	2 L UC	N/A	52 (95% CI 41, 62)	10.9 (95% CI 5.78, NR)	14.7 (95% CI 10.5, 18.2)	5.8 (95% CI 5.0, 8.3)	[37]
EV-301 Phase III NCT03474107	EV vs. Chemo.	2 L la/mUC	0.70 (95% CI 0.56, 0.89)	40.6 vs. 17.9	7.39 vs. 8.11	12.88 vs. 8.97	5.55 vs. 3.71	[17]
EV-103/KEYNOTE-869 Cohort K Phase Ib/II NCT03288545	EV +/− pembro.	1 L la/mUC	N/A	64.5 vs. 45.2	Not reached vs. 13.2	22.3 vs. 21.7	Not reached vs. 8	[59,60]
516-003 trial Phase II NCT03606174	EV + sitravatinib + pembro.?	2 L la/mUC	N/A	25	11.8	10.8	4.0	[61]
EV-302 Phase III NCT04223856	EV + pembro. vs. gemcitabine + platinum	1 L la/mUC	0.45 (95% CI 0.38, 0.54)	67.7 vs. 44.4	Not reached vs. 7.0	31.5 vs. 16.1	12.5 vs. 6.3	[27]
Phase IINCT06356155	EV +/− pembro. as neoadjuvant	Not yet recruiting	N/A	N/A	N/A	N/A	N/A	ClinicalTrials.com
Phase I/II NCT05845814	EV + pembro. plus investigational agents (anti-LAG-3 favezelimab or anti-Tight vibostolimab) Vs. EV + pembro.	Active, not recruiting	N/A	N/A	N/A	N/A	N/A	ClinicalTrials.com

Abbreviation: Chemo. = chemotherapy; EV = enfortumab vedotin; pembro. = pembrolizumab N/A: not available; +/− = with/without; * data for metastatic urothelial carcinoma.

**Table 2 cancers-16-03071-t002:** Clinical trials of pembrolizumab alone or in combination with chemotherapy or ADCs to treat urothelial cancer.

Study	Regimen	Inclusion Criteria	HR (mPFS)	ORR (months)	DoR (months)	mOS (months)	mPFS (months)	Citation
KEYNOTE-052 Phase II NCT02335424	Pembrolizumab	1 L la/mUC	N/A	24	NR (95% CI 9, NR)	N/A	N/A	[75]
KEYNOTE-052 Phase II NCT02335424 (long-term follow up)	Pembrolizumab	1 L la/mUC	N/A	28.6 (95% CI 24.1, 33.5)	30.1 (95% CI 18.1, NR)	11.3 (95% CI 9.7, 13.1)	2.2	[76]
KEYNOTE-045 Phase III NCT02256436 (two years follow-up)	Pembrolizumab vs. chemotherapy	2 L mUC	N/A	21.1 vs. 11.0%	Not reached vs. 4.4	10.3 vs. 7.4	2.1 vs. 3.3	[77]
KEYNOTE-057	Pembrolizumab	High-risk, non-muscle invasive UC	N/A	41 CR	N/A	N/A	N/A	[68]
KEYNOTE-361 Phase III NCT02853305	Pembrolizumab + chemo vs. pembrolizumab vs. chemo	1 L mUC	0.78 (95% CI 0.65, 0.93) (excludes monotherapy)	54.7 vs. 30.3 vs. 44.9	8.5 vs. 28.2 vs. 6.2	17.0 vs. 15.6 vs. 14.3	8.3 vs. 7.1 (excludes monotherapy)	[69]
Phase I/II NCT02437370	Pembrolizumab + either docetaxel or gemcitabine	2 L mUC	N/A	44 vs. 45	N/A	N/A	13.3 vs. 5.9	[78]
PEANUT Phase II NCT03464734	Pembrolizumab + nab-paclitaxel	≥2 L mUC	N/A	38.6 (95% CI 27, 51)	Not reached	Not reached (media follow-up 8.9)	5.9 (95% CI 3.1, 11.5)	[79]
Phase II NCT02581982	Pembrolizumab + paclitaxel	2 L mUC	N/A	33	N/A	11.7 months (95% CI 8.7, NR)	6 PFS 46.8	[80]
TROPHY U-01 cohort 3 Phase II NCT03547973	Pembrolizumab + sacituzumab govitecan	mUC after platinum	N/A	41 (95% CI, 26.3 to 57.9; 20% complete response rate)	11.1 (95% CI 4.8, NE)	12.7 (10.7, NE)	5.3 (95% CI 3.4–10.2)	[81]
RC48G001 Phase I/II NCT04879329	Disitamab vedotin +/− pembrolizumab	HER2+ 1 L mUC	Not recruiting	N/A	N/A	N/A	N/A	ClinicalTrials.gov
Phase III NCT05911295	Disitamab vedotin + pembrolizumab vs. chemotherapy	HER2+ 1 L la/mUC	Not recruiting	N/A	N/A	N/A	N/A	ClinicalTrials.gov

Abbreviation: Chemo. = chemotherapy; pembro. = pembrolizumab; N/A: not available; NR = not reached; NE = not estimable.

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
