# Peer review of "Mechanistic Insights into the Successful Development of Combination Therapy of Enfortumab Vedotin and Pembrolizumab for the Treatment of Locally Advanced or Metastatic Urothelial Cancer"

_cancers, 2024, doi:10.3390/cancers16173071_

Round 1
Reviewer 1 Report
Comments and Suggestions for Authors
Manuscript entitled "Mechanistic Insights into the Successful Development of Combination Therapy of Enfortumab Vedotin and Pembrolizumab for the Treatment of Locally Advanced or Metastatic Urothelial Cancer"
Major issues:
1. The authors described more clinical trial issues and less mechanistic Insights. I would suggest the authors to have more descriptions regarding cellular and molecular mechanisms.
2. The authors should have a picture summarizing the various molecular mechanisms regarding the treatment of UC in different settings.
3. The authors should summarize the expression/positive status of the key biomarkers in UC.
4. The authors should summarize the response rate in various clinical trials.
Comments on the Quality of English LanguageAcceptable
Author Response
- The authors described more clinical trial issues and less mechanistic Insights. I would suggest the authors to have more descriptions regarding cellular and molecular mechanisms.
As recommended, we discussed more on cellular and molecular mechanisms throughout the manuscript.
- The authors should have a picture summarizing the various molecular mechanisms regarding the treatment of UC in different settings.
As recommended, we added two summary figures (Figures 1 and 3) to summarize the major insights of this review paper.
- The authors should summarize the expression/positive status of the key biomarkers in UC.
We added one paragraph to discuss the potential markers for immunotherapy in UC and provided most recent references related to this topic.
- The authors should summarize the response rate in various clinical trials.
The response rates are summarized in Table 1 and Table 2.
Reviewer 2 Report
Comments and Suggestions for Authors
The article is well written and interesting, however I suggest making the following changes:
1) Eliminate (or drastically reduce) the paragraph entitled "The history of developing Enfortumab...": it is too verbose and essentially not very useful.
2) Explore the role of potential biomarkers of response to immunotherapy, perhaps creating a specific paragraph. An example of this could be the following articles:
doi: 10.1007/s40259-016-0176-3
doi: 10.1001/jamanetworkopen.2024.1215
3) In the context of the mechanisms of resistance to immunotherapy, it would be necessary to also mention the potential role of PD-L2 overexpression, which in recent years has become an increasingly topic of interest
4) It would be useful to delve deeper into the different type of response to immunotherapy based on the Robertson's classification of urothelial tumors, which the authors only briefly mention throughout the text
Author Response
1) Eliminate (or drastically reduce) the paragraph entitled "The history of developing Enfortumab...": it is too verbose and essentially not very useful.
We reduced the length of this section. However, we consider that this section helps the readers understand the molecular and cellular mechanisms of anti-Nectin-4 ADCs such as expression pattern and functions of Nextin-4.
2) Explore the role of potential biomarkers of response to immunotherapy, perhaps creating a specific paragraph. An example of this could be the following articles:
doi: 10.1007/s40259-016-0176-3
doi: 10.1001/jamanetworkopen.2024.1215
We reorganized the structure of the paper and performed in-depth discussion of the potential markers for immunotherapy. More relevant references are also cited to help readers to understand the mechanistic driven biomarker discovery.
3) In the context of the mechanisms of resistance to immunotherapy, it would be necessary to also mention the potential role of PD-L2 overexpression, which in recent years has become an increasingly topic of interest.
We agree with the reviewer’s comment that PD-L2 also plays important roles in resistance to immunotherapy. We addressed the role of PD-L2 in immunotherapy resistance in the revised manuscript.
4) It would be useful to delve deeper into the different type of response to immunotherapy based on the Robertson's classification of urothelial tumors, which the authors only briefly mention throughout the text.
We agree with the reviewer’s comments and further discussed on the responses to immunotherapy based on the Robertson's classification of urothelial tumors. However, due to space limit and the complexity of biomarker identification, we did not extensively discuss this issue. Instead, we added more references to help readers understand biomarker identification and validation of immune therapy in UC and other types of cancer.
Reviewer 3 Report
Comments and Suggestions for Authors
This paper explores the mechanistic insights behind the successful development of the combination therapy of enfortumab vedotin (EV), an antibody-drug conjugate (ADC), and pembrolizumab, an immune checkpoint inhibitor, for the treatment of locally advanced or metastatic urothelial carcinoma (la/mUC). The paper focuses on dissecting the biological mechanisms that contribute to the superior efficacy of this combination therapy compared to standard treatments.
The paper is well-written and provides valuable insights into a novel therapeutic approach for la/mUC. However, several areas could benefit from further refinement:
1. Clarification of Mechanistic Interpretations:
Some of the mechanistic explanations are vague and may lead to misunderstandings. Providing more detailed and precise explanations of the molecular mechanisms would enhance clarity.
2. Consistency of Data:
There are instances where results from different studies or data sources appear inconsistent, particularly regarding the relationship between Nectin-4 expression and therapeutic efficacy. It is important to emphasize the consistency of the data or explain why discrepancies might have occurred.
3. Improvement of Figures and Tables:
Some figures and tables do not effectively complement the text. Consider enhancing the visual representation of the mechanisms discussed to make them more accessible and easier to understand.
4. Strengthening the Conclusion:
The conclusion could be more impactful by emphasizing the key insights gained from the study and clearly outlining the implications for future research and clinical applications. This would leave a stronger impression on the reader.
Author Response
- Clarification of Mechanistic Interpretations:
Some of the mechanistic explanations are vague and may lead to misunderstandings. Providing more detailed and precise explanations of the molecular mechanisms would enhance clarity.
We restructured several portions of the manuscript and added two more Figures to help readers understand the mechanism of these therapies.
- Consistency of Data:
There are instances where results from different studies or data sources appear inconsistent, particularly regarding the relationship between Nectin-4 expression and therapeutic efficacy. It is important to emphasize the consistency of the data or explain why discrepancies might have occurred.
We restructured the paper and discussed in detail the potential role of Nectin-4 as a biomarker for therapeutic efficacy.
- Improvement of Figures and Tables:
Some figures and tables do not effectively complement the text. Consider enhancing the visual representation of the mechanisms discussed to make them more accessible and easier to understand.
Two Tables presented in this review are summaries of clinical studies. To help the readers understand the mechanisms of combination therapy and resistance to ICI therapy, we added two summarized graphic Figures (Figures 1 and 3).
- Strengthening the Conclusion:
The conclusion could be more impactful by emphasizing the key insights gained from the study and clearly outlining the implications for future research and clinical applications. This would leave a stronger impression on the reader.
We add two figures (Figures 1 and 3) to summarize the major insights of this review paper.
Round 2
Reviewer 1 Report
Comments and Suggestions for Authors
The revision is acceptable
Comments on the Quality of English LanguageAcceptable
Reviewer 2 Report
Comments and Suggestions for Authors
The authors have improved the manuscript, which can be published
Reviewer 3 Report
Comments and Suggestions for Authors
The revised manuscript should be accepted in present form. Nice work.